# The Dynamic Typology in the Development Process of Credit Union Movements

**Chien-Min Kang** [1,*] 📀, **Ming-Chieh Wang** [2] **and Lin Lin** [3] 📀

[1] The Ph.D. Program in Strategy and Development of Emerging Industries, College of Management, National Chi-Nan University, Nantou 54561, Taiwan
[2] Department of International Business Studies, National Chi-Nan University, Nantou 54561, Taiwan; mcwang@ncnu.edu.tw
[3] Newhuadu Business School, Min-Jiang University, Fuzhou 350108, China; linlin.ncnu@gmail.com
[*] Correspondence: arickkang@gmail.com; Tel.: +886-4-22917272

**Abstract:** The aim of this paper is to find a dynamic framework of analysis of credit union movements by grouping credit unions into different category types. Within the heterogeneous reality of the worldwide credit union movement, the typology provides a clearer understanding of the dynamics of change and development. We use panel-ordered logistic regression to find the key covariates of influence when analyzing the original typology to add further explanation of the development of credit union movements. By using transnational research, we revisit each of the three categories of the original typology to re-evaluate and expand upon the relevance of this particular model. We also include the elements of economy, society, education, and culture in other countries in this research. Our findings suggest that the stage of development of the credit union movement depends on the variables of asset scale, financial crisis, legislative framework, economy, society, and culture of the country. In addition, they indicate that the penetration rate of the credit union movement depends on the asset scale, loan ratio, credit union growth, financial crisis, economy, society, education, and culture of the country. This lends support to the recognition of the diversity of the credit unions' development.

**Keywords:** credit unions; ordered logistic regression; typology

## 1. Introduction

The research of Ferguson and McKillop (1997b) aims to offer a detailed consideration of the major determinants that carry forward the credit union movement of a particular country from one stage of development to another, using a typology as a starting point. The typology provides a useful tool for analyzing this dynamic category by offering a classification of credit union development. We try to find the key covariates of influence when analyzing the original typology to add further explanation of the development of the credit union movement. The applicability of the typology to research issues in cooperative forms of organization is also an important element of the research armory (Levi 1997).

Rostow (1971) distinguished five basic stages of economic growth experienced by societies as they change from a pre-industrial state to full economic maturity. The growth of the national income keeps pace with the stage of economic development. In addition to the changes in economic and political structure in the traditional stage, Rostow found that the value changed due to the expansion and improvement of education. Huntington (1971) noted that the process of modernization is considered to entail evolution, complexity, systematization, globalization, stage, homogeneity, inevitability, and progress. The population type is affected by changes in demographic structure. The population development is one important element of social development.

This paper concentrates on the development of credit unions rather than credit and savings societies and cooperative banks. In the worldwide credit union movement, there

are major differences in the development of particular countries. In the typology developed by Ferguson and McKillop (1997b), three discrete growth stages are identified: nascent, transition, and mature. Credit union movements in specific locations will potentially move from the beginning period through growth to full development as a result of the growth in their assets and members. The typology describes the development locus for credit unions, and it is a dynamic model within any one credit union movement at a given period of the same stage. The dynamic models and examinations are very important in economics and finance. As an example, Inci and Lu (2004) provide a thorough investigation of the dynamic relationship between exchange rates and interest rates internationally.

This quantitative research uses the dataset provided by the World Council of Credit Unions (WOCCU). In addition, data were collected from 60,645 credit unions in 109 countries on 6 continents around the world. By using the findings of this transnational research, the purpose of this paper is to revisit each of the three categories of the original typology to re-evaluate and expand upon the relevance of this particular model. Chang and Yu (2002), when constructing the national development index and ranking in 101 countries, found that their research results showed that economic indicators, social indicators, cultural indicators, educational indicators, and comprehensive indicators all have high reliability. McKillop and Wilson (2011) note that there is a great diversity within the credit union movement across these countries. This reflects the various economic, historic, and cultural contexts within which credit unions operate, and traces the evolution of the credit union movement. It also examines credit unions' objectives, and considers issues relating to efficiency, technology adoption, product diversification, merger, failure, and demutualization. The regulatory environment within which credit unions operate is also explored under the themes of interest rate regulation, common bond requirements, taxation, deposit insurance, and capital regulation. We include the elements of economy, society, education, and culture in different countries in this research.

The U.S. has progressed to its present position as a mature movement in spite of not being the headstream of credit unions. Ecuador provides a good example of a transition movement, although it may be viewed as unstable in the sense of being on the verge of mature status. The Dominican Republic is also an example of a movement in transition, although unlike Ecuador it is in the embryonic stage of development. Rwanda and Myanmar are in the nascent stage, and are characterized by an expanding number of small asset-scale credit unions. Concentrating on the variability of each country during the different stages gives a focus to this study of the dynamics of credit union development. As a starting point, it is useful to briefly summarize the basic features of each stage within the typology prior to a more extensive discussion of the forces that facilitate or hinder progression from one to another.

The rest of this paper is structured as follows: Section 2 provides a literature review of the credit union movement and its stages of development. Section 3 presents the data and model specifications. In Section 4, we discuss the empirical evidence and results. Finally, the last Section 5 presents the conclusions.

## 2. Literature Review

### 2.1. The Credit Union Movement

Credit unions are not-for-profit financial cooperatives. Each credit union is governed by its members, who elect unpaid volunteer officers and directors from within the membership. Voting is on a one-member-one-vote basis, regardless of the size of each member's financial stake (Goddard et al. 2009).

At the end of 2020 there were 86,451 credit unions in 108 countries all over the world, with a membership of 375 million, total assets of USD 3208 billion, and population penetration of 12.18%. There has been a rapid growth in the credit union movement over the past decade. Africa dominates in terms of 40,570 credit unions (46% of worldwide numbers), but only accounts for 11% of worldwide membership and 0.5% of worldwide assets. The population penetration rate for Africa is only 14.34%. The penetration rate

in North America is 50.48%. These rates seem to have significant heterogeneity across countries. For example, the penetration rate in Ireland is 111%, compared to 4.82% for Great Britain. The reserve-to-asset ratio is 8.2% across all countries, and ranges from 4.6% for Asia to 15.9% for Latin America by region. The World Council of Credit Unions (WOCCU) (2016a) suggests that the effective financial structure is institutional capital, as a percentage of reserves should be greater than or equal to 10% for a credit union. The loan-to-asset ratio ranges from 33.6% in Europe to 78.3% in Africa. The World Council of Credit Unions (WOCCU) (2016a) suggests that the effective financial structure as a loan-to-asset ratio is between 70 and 80% for a credit union.

In recent years, the assets and membership of credit unions have grown, but the number has declined through consolidation. As credit unions have become larger and more sophisticated, there has been a gradual shift away from using volunteers for day-to-day operational needs, and towards salaried employees. Credit unions serve a membership defined theoretically by a common bond (Goddard et al. 2002, 2008). This common bond might restrict membership to members of a local community, employees of a particular firm, or individuals with some other organizational affiliation.

Growth in membership has also been accompanied by product diversification, particularly in the case of the larger credit unions (Goddard et al. 2008). Many credit unions provide an array of retail financial services similar to those of banks and of savings and loan associations. In addition, credit unions may also offer interest-bearing business checking accounts and commercial loans, agricultural loans, and venture capital loans. Credit unions may also deal in investment products such as bankers' acceptances, cash-forward agreements, and reverse-purchase transactions. These product offerings have further blurred the lines of demarcation between credit unions and mainstream financial services providers (Tokle and Tokle 2000; Feinberg 2001; Feinberg and Rahman 2001; Schmid 2006).

### 2.2. Stages of Credit Union Development

Ferguson and McKillop (1997a, 1997b, 2000) use an organizational life-cycle methodology to partition credit unions into distinct growth phases. These phases include nascent (formative), transition, and mature. Credit unions positioned within each of these stages can be characterized by various financial and organizational attributes. The authors suggest that credit union movements at a nascent stage of development tend to have a small asset size, high levels of structural and conduct regulation, a tight common bond, a heavy reliance on volunteers, and provide basic savings and loans products. Transition movements are characterized by large asset size, evolving regulatory and supervisory frameworks, fewer common bond restrictions, higher levels of product diversification, developed professional trade associations, less reliance on volunteers, developed central services, and a greater emphasis on growth and efficiency. Mature movements have large asset size, have undergone structural and conduct deregulation accompanied by increased prudential regulation, a loose common bond, diversified product portfolios, professionalization of senior management, centralized services, adoption of electronic technologies, and a deposit insurance scheme. Goddard and Wilson (2005) applied this typology to credit unions within the United States, predicting that the cost function is a function of the developmental stage, since an institution in the earlier stages will use more volunteers, and be more likely to receive subsidies from its sponsor organization. Credit unions at a later stage of development might be run more professionally, with a management structure consisting of paid employees. This typology may be roughly correlated with size, but not directly. Table 1 shows the three growth stages identified for credit union movements. The main attributes are documented for each stage.

**Table 1.** Stages of credit union development.

| Stage | Attributes |
|---|---|
| Nascent Movement | Small asset size<br>Highly regulated<br>Tight common bond<br>Strong emphasis on voluntarism<br>Serve weak sections of society<br>Single savings and loans products<br>Requires sponsorship from wider credit union movement to take root<br>High commitment to traditional self-help ideals |
| Transition Movement | Large asset size<br>Shifts in regulatory framework<br>Adjustments to common bond<br>Shifts towards greater product diversification<br>Emphasis on growth and efficiency<br>Weakening of reliance on voluntarism<br>Recognition of the need for greater effectiveness and professionalism of trade bodies<br>Development of central services |
| Mature Movement | Large asset size<br>Deregulation<br>Loose common bond<br>Competitive environment<br>Electronic technology environment<br>Well-organized, progressive trade bodies<br>Professionalization of management<br>Well-developed central services<br>Diversification of products and services<br>Products and services based on market rate structures<br>Emphasis upon economic viability and long term sustainability<br>Rigorous financial management of operations<br>Well-functioning deposit insurance mechanism |

Note: the selection of stages and attributes was as described by Alexander Sibbald et al. (2002).

While it is important to understand the key characteristics of each stage, it is of more significance to understand the interaction of factors that create advancement between the different developmental stages. Sibbald et al. (2002) conceived a development framework to analyze credit union industries, where explicit consideration is given to credit union industries in four countries: Great Britain, Ireland, New Zealand, and the United States. In progression between stages, the analysis considers the influence of factors such as situational leadership, the complexion of trade associations, professionalization, regulatory and legislative initiatives, and technology. The analysis concluded that while there was a substantial commonality of experience, there were also significant differences in the impact of these factors. This consequently encouraged the recognition of the existence of "a variety of the species" with respect to credit union development.

The key factor determining whether a credit union movement is nascent, or has advanced to a higher stage of development, is leadership. Credit union leaders have tended to have charisma, vision, commitment, deeply held religious beliefs emphasizing service, and a strong missionary zeal. How credit unions integrate themselves into trade associations is another factor that contributes to the development from one stage to another. As umbrella organizations for credit unions, trade associations represent their interests to governments and government agencies, including regulators. They also provide training, information technology, marketing advice, and other services to members. The early development of credit unions in the nascent stage inevitably means that volunteer staff run the movement. However, growth and the complexity of products and services means that the control of directors diminishes as paid professional staff take over the operation

of credit unions in a highly technological world. The downside of this professionalism is that the highly personal nature of credit unions diminishes (Black and Dugger 1981). First-generation volunteers are totally committed and steeped in tradition, while next-generation management tends to be recruited from outside the movement. Since paid managers see themselves as professionals, there is a phenomenon whereby they might not consider themselves totally accountable to a volunteer board of directors. It is possible that the introduction of innovation and technology has tended to give professional staff greater power, as directors have tended to leave much of the decision making to them.

Thomas and Balloch (1994) noted that credit unions in the nascent stage are operated by volunteers, and require sponsorship and various kinds of help from local authorities and other credit unions. The Commission on Credit Unions (2012) uses the organizational life-cycle typology to distinguish country-specific movements into one of the three growth phases. In Table 2, this typology is used to classify credit union movements in those countries with the data updated to 2015. In the nascent stage, they have a small asset size, are highly regulated, and have a tight common bond, with a strong commitment to traditional self-help ideals. Examples of credit union movements at this stage are found not only in Rwanda and Mali in Africa, but also in the Caribbean (particularly Haiti) and in Southeast Asia (particularly in Myanmar). However, these movements might develop through either establishing larger credit unions or creating many new unions. Once a particular critical size has been reached, the movement advances to the second (transition) stage of development, which represents the stage before maturity.

**Table 2.** Geographic location of movement types.

| Country | Number of Credit Unions | Membership | Assets (million USD) | Penetration [1] (%) |
|---|---|---|---|---|
| Mature Credit Unions | | | | |
| United States | 6100 | 103,709,631 | 1,215,943 | 48.8 |
| Canada | 695 | 10,348,048 | 249,276 | 44.1 |
| Australia | 91 | 4,100,000 | 70,706 | 27.0 |
| India | 2705 | 21,060,430 | 60,450 | 2.6 |
| Thailand | 2277 | 4,078,311 | 57,101 | 8.2 |
| Korea | 910 | 5,752,000 | 56,111 | 16.0 |
| Brazil | 582 | 6,339,462 | 28,239 | 4.5 |
| Ireland | 421 | 3,400,000 | 16,816 | 77.0 |
| Transition Credit Unions | | | | |
| Kenya | 5769 | 5,432,009 | 5355 | 21.3 |
| Hong Kong | 41 | 86,558 | 1631 | 1.7 |
| ROC Taiwan | 340 | 217,909 | 844 | 1.3 |
| Sri Lanka | 8423 | 1,039,458 | 83 | 7.2 |
| Singapore | 22 | 103,444 | 671 | 2.3 |
| Indonesia | 912 | 2,640,692 | 1890 | 1.5 |
| Philippines | 1649 | 4,091,059 | 2646 | 6.6 |
| Vietnam | 1148 | 2,097,584 | 2705 | 3.2 |
| Colombia | 193 | 2,408,000 | 4700 | 8.5 |
| Ecuador | 900 | 4,758,802 | 8100 | 46.2 |
| Mexico | 142 | 5,140,944 | 5264 | 6.4 |

**Table 2.** *Cont.*

| Country | Number of Credit Unions | Membership | Assets (million USD) | Penetration [1] (%) |
|---|---|---|---|---|
| | | Transition Credit Unions | | |
| New Zealand | 13 | 180,916 | 673 | 6.2 |
| Dominican Republic | 15 | 645,331 | 910 | 9.4 |
| Jamaica | 34 | 999,416 | 748 | 52.8 |
| Trinidad & Tobago | 128 | 651,388 | 1982 | 75.3 |
| Great Britain | 342 | 1,269,345 | 2028 | 3.1 |
| Poland | 48 | 2,072,598 | 3172 | 7.7 |
| | | Nascent Credit Unions | | |
| Rwanda | 416 | 1,607,560 | 137 | 22.8 |
| Mali | 70 | 911,794 | 116 | 10.8 |
| Myanmar | 2228 | 388,258 | 37 | 1.0 |
| Moldova | 295 | 126,453 | 26 | 5.1 |
| Mongolia | 253 | 39,146 | 49 | 1.9 |
| Guyana | 28 | 34,212 | 29 | 7.0 |
| Romania | 19 | 68,103 | 61 | 0.5 |

Source of data: World Council of Credit Unions (World Council of Credit Unions 2016b). Data relate to the end of 2015. http://www.woccu.org/publications/statreport (accessed on 24 November 2017). [1] The penetration rate is calculated by dividing the total number of reported credit union members by the economically active population.

In the transition stage, credit unions need to achieve a larger asset scale to emphasize growth in membership as the credit union's objectives. They inevitably hire paid employees, whether full-time or part-time. They also have professional management, but retain volunteer directors. They may have greater product diversification, and may offer a range of services such as life-saving and loan-protection insurance, member benefits, or special savings and loans facilities. Credit union membership adjusts to contain a social diversity of common bonds, including the unemployed, low-income households, and even middle-class families. The credit union movements of Kenya, Ecuador, and Mexico provide a good example of the transition stage, as they are currently on the edge of entry to the mature stage.

The final stage of development—maturity—has well-developed business-like banking. Examples include the credit union movements found in the US, Canada, and Australia. The features of the mature stage are large asset and member size, with well-organized and progressive trade bodies through a merger process. The underlying legislative environment is relaxed considerably to encourage competition with other financial organizations. This legislative change presents the movement with extensive new opportunities for growth by permitting credit unions to become national rather than provincial in scope and dimension. It also allows credit unions to merge. There is also professionalization of management, as well as diversification of products and services, which are often driven by the electronic technology—including, for instance, debit, credit, and smart cards, checkbook facilities, mortgages, and Internet services. However, even at the mature stage, the movement is still characterized by the superiority of volunteer directors (Sibbald et al. 1999).

### 3. Methods

*3.1. Variables Definition*

3.1.1. Dependent Variables

This study examines the development of credit union movements worldwide, and their influence on the classification of credit unions. We use three discrete growth stages and penetration rate as the dependent variables. According to various financial and organizational

attributes of the stages of credit unions' development (Ferguson and McKillop 1997a, 1997b, 2000; Sibbald et al. 2002; Commission on Credit Unions 2012), we use the attribute of asset size to define the stages of credit unions. The asset size data of each country were obtained from the statistical report available on the website of the World Council of Credit Unions (World Council of Credit Unions 2016b; http://www.woccu.org/publications/statreport, accessed on 24 November 2017).

### 3.1.2. Independent Variables

Previous literature shows that the asset scale, loan ratio, and deregulation epitomize the relationship between the development of credit union movements and their stage of classification. Additionally, the asset scale is affected by the growth of credit unions and their membership. We chose asset scale, loan ratio, credit union growth, member growth, financial crisis, and deregulation as the independent variables (Thomas and Balloch 1994; Ferguson and McKillop 1997a, 1997b, 2000; Sibbald et al. 2002).

### 3.1.3. Proxy Variable

We use gross domestic product (GDP), urban population ratio, and non-agricultural employment rate as the proxy variables of economic indicators; health expenditure ratio, birth control rate, and life expectancy as the proxy variables of social indicators; government expenditure on education and adult literacy rate as the proxy variables of educational indicators; and R&D expenditure ratio and Internet users as the proxy variables of cultural indicators (Huntington 1971; Rostow 1971; Morris 1979; Hettne 1990; United Nations Development Programme(UNDP) 1995; Chang and Yu 2002).

### 3.1.4. Research Variables

This research consists of 6 independent variables and 10 proxy variables. Table 3 shows the definitions of dependent variable, independent variable, and proxy variable used in this research.

**Table 3.** Covariates of credit unions used in this research (1995~2015).

| Covariates | Description |
|---|---|
| **Dependent Variable** | |
| STAGE | Nascent: 1; transition: 2; mature: 3 |
| PENETRATION | Total number of reported credit union members/the economically active population aged 15–64 years old. |
| **Independent Variable** | |
| AVG_ASSET (C1) | Log-average assets of credit union |
| LOAN_RATIO (C2) | Total loans/total assets |
| CU_GRO (C3) | (CU No. of year $t$—CU No. of year $t-1$)/CU No. of year $t-1$ |
| MEM_GRO (C4) | (Members of year $t$—members of year $t-1$)/members of year $t-1$ |
| CRISIS (C5) | The time before the financial crisis of 2008: 0; the time after the financial crisis of 2008: 1 |
| DEREGULATION (C6) | The time before the WOCCU declared the importance of legislative frameworks in 2002: 0; the time after the WOCCU declared the importance of legislative frameworks in 2002: 1 |
| **Proxy Variable** | |
| GDP (C7) | Gross domestic product divided by midyear population. |
| URBAN_RATIO (C8) | Urban population refers to people living in urban areas |
| EMP_NONAGR_RATIO (C9) | Employment in non-agriculture to population ratio is the proportion of a country's population that is employed. People aged 15 years and older are generally considered to be the working-age population. |

**Table 3.** *Cont.*

| Covariates | Description |
|---|---|
| Proxy Variable | |
| HEALTH_EXP_RATIO (C10) | Total health expenditure is the sum of public and private health expenditure. |
| BIRTH_CON_RATE (C11) | Reciprocal of total fertility rate. Total fertility rate represents the number of children that would be born to a woman if she were to live to the end of her childbearing years and bear children in accordance with age-specific fertility rates of the specified year. |
| LIFE_EXP (C12) | Life expectancy at birth indicates the number of years a newborn infant would live if prevailing patterns of mortality at the time of their birth were to stay the same throughout their life. |
| GOVEXP_EDU_RATIO (C13) | General government expenditure on education (current, capital, and transfers) is expressed as a percentage of GDP. |
| ADULT_LIT_RATE (C14) | Percentage of the population aged 15 and above who can, with understanding, read and write a short, simple statement about their everyday life. |
| RD_EXP_RATIO (C15) | Expenditure for research and development includes current and capital expenditure on creative work undertaken systematically to increase knowledge, including knowledge of humanity, culture, and society, and the use of knowledge for new applications. |
| INTERNET_USERS (C16) | Internet users are individuals who have used the Internet in the last 12 months. The Internet can be used via a computer, mobile phone, personal digital assistant, games machine, digital TV, etc. |

Note: the selection and definition of covariates was as described by Huntington (1971), Rostow (1971), Morris (1979), Hettne (1990), the United Nations Development Programme(UNDP) (1995), Alexander Sibbald et al. (2002), and the World Bank.

### 3.2. Research Design

#### 3.2.1. Data Source

The annual financial data for the credit unions were obtained from the statistical report available on the website of the World Council of Credit Unions (WOCCU; http://www.woccu.org/publications/statreport, accessed on 24 November 2017). The data of the proxy variables for economic, social, educational, and cultural indicators were collected from the World Bank website (https://data.worldbank.org, accessed on 24 November 2017).

#### 3.2.2. Data

This research concentrates on the relationship between the development of credit union movements worldwide and their stage of classification. We also focus on the key element of how to enhance the penetration rate in different countries. We covered a 20-year period from 1995 to 2015. The data of 1860 credit unions was obtained from 109 countries. The data of 1659 credit unions were left finally. The data were from 1996 to 2015, excluding missing data and the data of 1995 as the base period. The results reported in this paper are based on this dataset.

#### 3.2.3. Data Analysis Method

We used an unbalanced panel data with cross-section and time-series characteristics. The statistical software STATA was used in this research. The regression analysis models were an ordered logistic model and a fixed-effects model.

#### 3.2.4. Ordered Logistic Model

Ordered logistic models are used to estimate relationships between an ordinal dependent variable and a set of independent variables. An ordinal variable is a variable that is categorical and ordered—for instance, "nascent", "transition", and "mature", which might indicate the current status of the credit unions in each country. The actual values taken on by the dependent variable are irrelevant, although larger values are assumed to correspond to "higher" outcomes. The conditional distribution of the dependent variable given the

random effects is assumed to be multinomial, with the probability of success determined by the logistic cumulative distribution function.

Maximum likelihood of the random-effects model:

$$\Pr (y_{it} > k \mid \kappa, x_{it}, v_i) = H(x_{it}\beta + v_i - \kappa_k) \tag{1}$$

for $i = 1, \ldots, n$ panels, where $t = 1, \ldots, n_i$, and $v_i$ are independent and identically distributed $N(0, \sigma_v^2)$, and $\kappa$ is a set of cut-points $\kappa_1, \kappa_2, \ldots, \kappa_{k-1}$, where $K$ is the number of possible outcomes, while $H(\cdot)$ is the logistic cumulative distribution function.

From the above, we can derive the probability of observing outcome $k$ for response $y_{it}$ as follows:

$$
\begin{aligned}
P_{it}k \equiv \Pr (y_{it} > k \mid \kappa, x_{it}, v_i) &= \Pr (\kappa_{k-1} < x_{it}\beta + v_i + \in_{it} \leqq \kappa_k) \\
&= \Pr(\kappa_{k-1} - x_{it}\beta - v_i < \in_{it} \leqq \kappa_k - x_{it}\beta - v_i) \\
&= H(\kappa_k - x_{it}\beta - v_i) - H(\kappa_{k-1} - x_{it}\beta - v_i) \\
&= \frac{1}{1+\exp(-\kappa_k + x_{it}\beta + v_i)} - \frac{1}{1+\exp(-\kappa_{k-1} + x_{it}\beta + v_i)}
\end{aligned}
\tag{2}
$$

where $\kappa_0$ is taken as $-\infty$ and $\kappa_k$ is taken as $+\infty$. Here, $x_{it}$ does not contain a constant term, because its effect is absorbed into the cut-points.

We may also express this model in terms of a latent linear response, where observed ordinal responses $y_{it}$ are generated from the latent continuous responses, such that:

$$y_{it}^* = x_{it}\beta + v_i + \in_{it} \tag{3}$$

and:

$$
y_{it} = \begin{cases}
1 & \text{if} \quad y_{it}^* \leq \kappa_1 \\
2 & \text{if} \quad \kappa_1 < y_{it}^* \leq \kappa_2 \\
\vdots \\
\vdots \\
K & \text{if} \quad \kappa_{k-1} < y_{it}^*
\end{cases}
$$

The errors $\in_{it}$ are distributed as logistic, with mean zero and variance $\pi_{2/3}$, and are independent of $v_i$.

## 4. Results

When discussing which covariates of the model to select in order to estimate the stages of credit unions, this study explains the sample summary statistics for each of the covariates. The summary statistics are reported in Tables 4 and 5. Table 4 reports sample means, standard deviations, and correlation coefficients for the time-varying covariates of the model. To calculate and illustrate these summary statistics, the annual observations for each sample country from the period 1995 to 2015 are pooled.

Table 5 reports the mean of time-varying covariates among mature/transition/nascent-stage samples collected from 1995 to 2015. An increase in the mean of covariates from the nascent to mature stages during 1995–2015 is indicated for the average assets of the credit unions (AVG_ASSET), gross domestic product divided by midyear population (GDP), urban population (referring to people living in urban areas) (URBAN_RATIO), employment in non-agriculture–population ratio (EMP_NONAGR_RATIO), reciprocal of total fertility rate (BIRTH_CON_RATE), life expectancy at birth (LIFE_EXP), and Internet users (INTERNET_USERS), while a decrease is indicated for total loan/total assets (LOAN_RATIO) and member growth ratio (MEM_GRO).

**Table 4.** Summary statistics: time-varying covariates.

| Covariate | Obs | Mean | Std. Dev. | Minimum | Maximum |
|---|---|---|---|---|---|
| $AVG\_ASSET_{i,t}(Ln)$ | 1659 | 14.27 | 2.557 | 6.44 | 21.19 |
| $LOAN\_RATIO_{i,t}$ | 1659 | 0.676 | 0.173 | 0.014 | 0.99 |
| $CU\_GRO_{i,t}$ | 1659 | 0.161 | 1.801 | −0.997 | 44.00 |
| $MEM\_GRO_{i,t}$ | 1659 | 0.173 | 0.983 | −0.964 | 21.02 |
| $CRISIS_{i,t}$ | 1659 | 0.464 | 0.499 | 0 | 1 |
| $DEREGULATION_{i,t}$ | 1659 | 0.754 | 0.431 | 0 | 1 |
| $GDP_{i,t}$ | 1642 | 9597.3 | 14,595.35 | 149 | 93,606 |
| $URBAN\_RATIO_{i,t}$ | 1645 | 0.529 | 0.238 | 0.084 | 1.00 |
| $EMP\_NONAGR\_RATIO_{i,t}$ | 1480 | 0.408 | 0.09 | 0.088 | 0.597 |
| $HEALTH\_EXP\_RATIO_{i,t}$ | 1580 | 0.061 | 0.022 | 0 | 0.171 |
| $BIRTH\_CON\_RATE_{i,t}$ | 1565 | 0.433 | 0.184 | 0.132 | 1.11 |
| $LIFE\_EXP_{i,t}$ | 1545 | 68.95 | 8.852 | 35.66 | 83.98 |
| $GOVEXP\_EDU\_RATIO_{i,t}$ | 1297 | 0.044 | 0.015 | 0.011 | 0.1 |
| $ADULT\_LIT\_RATE_{i,t}$ | 928 | 0.828 | 0.179 | 0.15 | 1.00 |
| $RD\_EXP\_RATIO_{i,t}$ | 915 | 0.007 | 0.0076 | 0 | 0.043 |
| $INTERNET\_USERS_{i,t}$ | 1605 | 24.11 | 25.11 | 0 | 98.32 |
| $PENETRATION_{i,t}$ | 1659 | 0.166 | 0.285 | 0 | 2.88 |
| $STAGE_{i,t}$ | 1659 | 1.547 | 0.633 | 1 | 3 |

Source of data: 1. World Council of Credit Unions (WOCCU). http://www.woccu.org/publications/statreport (accessed on 24 November 2017). 2. World Bank website. https://data.worldbank.org (accessed on 24 November 2017). 3. This table reports summary statistics for all of the covariates used in the ordered logistic models and panel data models. The covariates' description follows in Table 3. 4. AVG_ASSET, LOAN_RATIO, CU_GRO, MEM_GRO, CRISIS, and DEREGULATION are the proxy variables of credit union movements (Sibbald et al. 2002). GDP, URBAN_RATIO, and EMP_NONAGR_RATIO are the proxy variables of economic indicators (Rostow 1971). HEALTH_EXP_RATIO, BIRTH_CON_RATE, and LIFE_EXP are the proxy variables of social indicators (Huntington 1971; Morris 1979). GOVEXP_EDU_RATIO, ADULT_LIT_RATE are the proxy variables of educational indicators (Rostow 1971; United Nations Development Programme(UNDP) 1995). RD_EXP_RATIO and INTERNET_USERS are the proxy variables of cultural indicators (Rostow 1971).

**Table 5.** The mean of time-varying covariates among mature/transition/nascent-stage samples.

| Covariate | Mature Stage | | | Transition Stage | | | Nascent Stage | | |
|---|---|---|---|---|---|---|---|---|---|
| | Mean | St. Dev. | Median | Mean | St. Dev. | Median | Mean | St. Dev. | Median |
| $AVG\_ASSET_{i,t}(Ln)$ | 17.69 | 1.39 | 17.55 | 15.22 | 2.09 | 15.51 | 13.07 | 2.24 | 13.16 |
| $LOAN\_RATIO_{i,t}$ | 0.657 | 0.147 | 0.657 | 0.655 | 0.177 | 0.69 | 0.694 | 0.171 | 0.721 |
| $CU\_GRO_{i,t}$ | −0.013 | 0.119 | −0.028 | 0.301 | 2.772 | 0 | 0.082 | 0.619 | 0 |
| $MEM\_GRO_{i,t}$ | 0.037 | 0.115 | 0.015 | 0.11 | 0.389 | 0.046 | 0.24 | 1.305 | 0.047 |
| $CRISIS_{i,t}$ | 0.504 | 0.502 | 1 | 0.599 | 0.49 | 1 | 0.357 | 0.479 | 0 |
| $DEREGULATION_{i,t}$ | 0.795 | 0.405 | 1 | 0.858 | 0.349 | 1 | 0.671 | 0.47 | 1 |
| $GDP_{i,t}$ | 28,161.1 | 19,353.1 | 24,155.8 | 9486.3 | 12,504.9 | 4274.9 | 6963.9 | 13,221.9 | 2640.1 |
| $URBAN\_RATIO_{i,t}$ | 0.718 | 0.184 | 0.801 | 0.565 | 0.241 | 0.595 | 0.475 | 0.223 | 0.46 |
| $EMP\_NONAGR\_RATIO_{i,t}$ | 0.467 | 0.069 | 0.479 | 0.414 | 0.09 | 0.415 | 0.393 | 0.088 | 0.402 |
| $HEALTH\_EXP\_RATIO_{i,t}$ | 0.086 | 0.035 | 0.083 | 0.059 | 0.016 | 0.059 | 0.059 | 0.021 | 0.058 |
| $BIRTH\_CON\_RATE_{i,t}$ | 0.58 | 0.125 | 0.555 | 0.437 | 0.181 | 0.409 | 0.407 | 0.183 | 0.387 |
| $LIFE\_EXP_{i,t}$ | 77.08 | 4.52 | 78.54 | 70.93 | 6.97 | 72.03 | 66.18 | 9.44 | 69.21 |

**Table 5.** *Cont.*

| Covariate | Mature Stage | | | Transition Stage | | | Nascent Stage | | |
|---|---|---|---|---|---|---|---|---|---|
| | Mean | St. Dev. | Median | Mean | St. Dev. | Median | Mean | St. Dev. | Median |
| GOVEXP_EDU_RATIO$_{i,t}$ | 0.048 | 0.007 | 0.049 | 0.044 | 0.014 | 0.044 | 0.043 | 0.017 | 0.042 |
| ADULT_LIT_RATE$_{i,t}$ | 0.871 | 0.126 | 0.926 | 0.833 | 0.175 | 0.908 | 0.819 | 0.185 | 0.875 |
| RD_EXP_RATIO$_{i,t}$ | 0.018 | 0.009 | 0.018 | 0.0051 | 0.005 | 0.003 | 0.0057 | 0.006 | 0.0037 |
| INTERNET_USERS$_{i,t}$ | 51.19 | 26.98 | 58.75 | 27.61 | 24.23 | 20.81 | 17.5 | 22.08 | 7 |
| PENETRATION$_{i,t}$ | 0.298 | 0.25 | 0.247 | 0.171 | 0.254 | 0.068 | 0.144 | 0.306 | 0.025 |
| N | | 127 | | | 654 | | | 878 | |
| Marginal Percentage | | 7.6% | | | 39.4% | | | 53% | |

*4.1. Estimated Results of Ordered Logistic Regression*

The relationship between asset size and the discrete growth stages is widely documented in the previous theoretical and empirical literature (Thomas and Balloch 1994; Ferguson and McKillop 1997a, 1997b, 2000; Sibbald et al. 2002). In the nascent stage, credit unions have small asset size, are highly regulated, and have a tight common bond, with strong commitment to traditional self-help ideals. In the transition stage, credit unions need to achieve larger asset size to emphasize growth in membership and legislation as the credit union's objectives. The features of the mature stage are large asset and member size, with well-organized and progressive trade bodies through a merger process. Accordingly, the relationship between asset scale and stages means that small-scale of credit unions is the main feature of the nascent stage, just as larger-scale institutions is in the mature stage. Table 6 shows a significant positive coefficient on AVG_ASSET in the ordered logistic regression, while Table 5 indicates a stable increase in the mean assets of credit unions throughout the sample period from the nascent stage to mature stage.

The estimated results of the ordered logistic regression are shown in Table 6. A positive coefficient on a particular variable implies that the probability of being at the mature stage relative to the nascent stage increases when that variable increases by one unit. Estimation I represents the results of the credit union movement. Estimation II represents the results of the credit union movement affected by economic, social, educational, and cultural indicators. Estimation III represents the results of the credit union movement affected by economic, social, educational, and cultural indicators in the context of the financial crisis of 2008. Estimation IV represents the results of the credit union movement affected by economic, social, educational, and cultural indicators around the time that the WOCCU declared the importance of legislative frameworks in 2002. Estimation V represents the results of the credit union movement affected by economic, social, educational, and cultural indicators, the financial crisis and the legislative framework declaration.

As shown in Table 6, the loan ratio, the growth ratio of credit unions, and the growth ratio of membership represent the development of credit unions in each country; countries with a high loans-to-assets ratio or a high growth ratio of membership might be common phenomena in the nascent stage. Thus, we would expect a negative coefficient on LOAN_RATIO and MEM_GRO in the ordered logistic regression, but only the negative coefficient on of MEM_GRO in Estimation I is significant. Alternatively, the positive coefficient on CU_GRO in Estimation I means that the mean value of CU_GRO in the transition stage is obviously higher than in the other two stages, according to Table 5.

**Table 6.** Estimated results of the ordered logistic regression.

| Estimation/Stage | I | II | III | IV | V |
|---|---|---|---|---|---|
| $AVG\_ASSET_{i,t}(Ln)$ | 2.00 *** (0.147) | 1.54 *** (0.276) | 1.47 *** (0.274) | 1.51 *** (0.285) | 1.44 *** (0.284) |
| $LOAN\_RATIO_{i,t}$ | −1.187 (0.898) | −0.376 (1.78) | −0.624 (1.78) | −0.312 (1.84) | −0.569 (1.84) |
| $CU\_GRO_{i,t}$ | 0.192 *** (0.059) | 0.063 (0.1) | 0.065 (0.102) | 0.076 (0.096) | 0.078 (0.097) |
| $MEM\_GRO_{i,t}$ | −0.709 ** (0.284) | −0.558 (0.357) | −0.511 (0.353) | −0.639 * (0.372) | −0.589 (0.371) |
| $CRISIS_{i,t}$ | | | 1.202 * (0.666) | | 1.203 * (0.682) |
| $DEREGULATION_{i,t}$ | | | | 2.162 * (1.27) | 2.097 * (1.26) |
| $GDP_{i,t}$ | | −0.00007 (0.0001) | −0.00009 (0.0001) | −0.00006 (0.0001) | −0.00007 (0.0001) |
| $URBAN\_RATIO_{i,t}$ | | −8.51 * (5.12) | −7.31 (5.09) | −8.9 * (5.39) | −7.61 (5.38) |
| $EMP\_NONAGR\_RATIO_{i,t}$ | | 5.51 (3.723) | 5.19 (3.68) | 5.25 (3.74) | 5.05 (3.71) |
| $HEALTH\_EXP\_RATIO_{i,t}$ | | −23.81 (26.9) | −28.32 (27.4) | −31.52 (28.3) | −35.53 (28.8) |
| $BIRTH\_CON\_RATE_{i,t}$ | | −7.713 * (4.37) | −6.885 (4.37) | −8.383 * (4.55) | −7.599 * (4.57) |
| $LIFE\_EXP_{i,t}$ | | 0.249 ** (0.099) | 0.244 ** (0.099) | 0.257 ** (0.101) | 0.253 ** (0.102) |
| $GOVEXP\_EDU\_RATIO_{i,t}$ | | 29.13 (28.95) | 28.14 (29.66) | 37.45 (30.27) | 37.01 (31.07) |
| $ADULT\_LIT\_RATE_{i,t}$ | | −1.49 (5.4) | −1.111 (5.4) | −0.562 (5.63) | −0.226 (5.7) |
| $RD\_EXP\_RATIO_{i,t}$ | | 364.52 ** (145.1) | 369.21 ** (144.7) | 382.65 ** (153.5) | 386.03 ** (153.5) |
| $INTERNET\_USERS_{i,t}$ | | 0.028 (0.024) | 0.003 (0.027) | 0.021 (0.025) | −0.004 (0.029) |
| Cut1 | 29.28 *** (2.42) | 31.19 *** (8.45) | 30.72 *** (8.41) | 33.43 *** (8.86) | 33.01 *** (8.88) |
| Cut2 | 41.16 *** (3.01) | 46.43 *** (9.66) | 45.91 *** (9.52) | 49.33 *** (10.21) | 48.81 *** (10.12) |
| Estimated Variance | 41.21 (12.98) | 26.89 (12.69) | 26.11 (12.1) | 29.71 (14.56) | 29.01 (14.04) |
| LR test | 1359.8 *** | 222.5 *** | 219.16 *** | 223.67 *** | 221.32 *** |

Note: 1. A positive coefficient on a particular variable implies that the probability of being at the mature stage relative to the nascent stage increases when that variable increases by one unit. Standard errors are in parentheses (*** estimated coefficient significantly different from zero, two-tailed test, 1% significance level; ** as above, 5% significance level; * as above, 10% significance level). 2. Estimation I represents the results of the credit union movement. Estimation II represents the results of the credit union movement affected by economic, social, educational, and cultural factors. Estimation III represents the results of the credit union movement affected by economic, social, educational, and cultural factors in the context of the financial crisis of 2008. Estimation IV represents the results of the credit union movement affected by economic, social, educational, and cultural factors around the time that the WOCCU declared the importance of legislative frameworks in 2002. Estimation V represents the results of the credit union movement affected by economic, social, educational, and cultural factors, the financial crisis and the legislative framework declaration.

For economic indicators, we use gross domestic product (GDP), urban population ratio (URBAN_RATIO), and non-agricultural employment rate (EMP_NONAGR_RATIO) as the proxy variables. Gross domestic product (GDP) does not have a significant relationship with the stages of development, indicating that GDP does not affect the latter. The ratio of urban population has significant negative relationship with the stages of development in Estimations II and IV, which means that the higher the urban population ratio (URBAN_RATIO), the lower the probability of being at the mature stage relative to the nascent stage. Finally, the non-agricultural employment rate (EMP_NONAGR_RATIO) presents an insignificant positive relationship with the stages of development, indicating that the non-agricultural employment rate does not affect the stages of development, as shown in Table 6.

This research uses health expenditure ratio, birth control rate, and life expectancy as the proxy variables of social indicators. The life expectancy (LIFE_EXP) shows a significant positive relationship with the stages of development in Estimations II, III, IV, and V: the longer life expectancy at birth, the higher the probability of being at the mature stage relative to the nascent stage. Birth control rate (BIRTH_CON_RATE) has a significant negative relationship with the stages of development in Estimations II, IV, and V: the lower the reciprocal of the total fertility rate, the higher the probability of being at the mature stage relative to the nascent stage. Finally, the health expenditure ratio (HEALTH_EXP_RATIO) presents an insignificant negative relationship with the stages of development, indicating that the health expenditure ratio does not affect the stages of development.

For educational indicators, we use government expenditure on education (GOVEXP_EDU_RATIO) and adult literacy rate (ADULT_LIT_RATE) as the proxy variables. Both GOVEXP_EDU_RATIO and ADULT_LIT_RATE have insignificant relationships with the stages of development, indicating that they do not affect the latter.

Finally, we use R&D expenditure ratio (RD_EXP_RATIO) and Internet users (INTERNET_USERS) as the proxy variables of cultural indicators. Only the R&D expenditure ratio shows a significant positive relationship with the stages of development, in Estimations II, III, IV, and V: the higher the R&D expenditure ratio, the higher the probability of being at the mature stage relative to the nascent stage. The number of Internet users (INTERNET_USERS) has no significant relationship with the stages of development, indicating that INTERNET_USERS has no effect on the stages of development. As we know, the financial crisis (CRISIS) of 2008 and the WOCCU declaration of the importance of legislative frameworks (DEREGULATION) in 2002 show significant positive relationships with the stages of development: the longer after the financial crisis of 2008 or the WOCCU declaration of the importance of legislative frameworks in 2002, the higher the probability of being at the mature stage relative to the nascent stage.IV

The estimated results of binary logistic regression are shown in Tables 7 and 8. A positive coefficient on a particular variable implies that the probability of being at the transition stage relative to nascent stage increases when that variable increases by one unit (Table 7). The estimated results for the transition stage relative to the mature stage are shown in Table 8. Estimation I represents the results of the credit union movement. Estimation II represents the results of the credit union movement affected by economic, social, educational, and cultural indicators. Estimation III represents the results of the credit union movement affected by economic, social, educational, and cultural indicators in the context of the financial crisis of 2008. Estimation IV represents the results of the credit union movement affected by economic, social, educational, and cultural indicators around the time that the WOCCU declared the importance of legislative frameworks in 2002. Estimation V represents the results of the credit union movement affected by economic, social, educational, and cultural indicators in the context of both the financial crisis and the legislative framework declaration.

**Table 7.** Estimated results of the binary logistic regression (from the nascent stage to the transition stage).

| Estimation/Stage | I | II | III | IV | V |
|---|---|---|---|---|---|
| Constant Term | −5.568 *** (−12.09) | −16.495 *** (−6.27) | −17.139 *** (−6.45) | −17.906 *** (−6.23) | −18.091 *** (−6.29) |
| AVG_ASSET$_{i,t}$(Ln) | 0.467 *** (15.83) | 0.949 *** (10.16) | 0.942 *** (10.08) | 0.945 *** (10.09) | 0.940 *** (10.04) |
| LOAN_RATIO$_{i,t}$ | −1.961 *** (−5.399) | −2.861 *** (−3.20) | −2.970 *** (−3.28) | −2.932 *** (−3.25) | −3.105 *** (−3.31) |
| CU_GRO$_{i,t}$ | 0.317 ** (2.49) | 0.102 (0.92) | 0.110 (0.96) | 0.117 (1.00) | 0.120 (1.01) |
| MEM_GRO$_{i,t}$ | −0.402 ** (−2.46) | −0.519 * (−1.75) | −0.484 * (−1.68) | −0.561 * (−1.87) | −0.517 * (−1.77) |
| CRISIS$_{i,t}$ | | | 0.736 ** (2.14) | | 0.673 * (1.92) |
| DEREGULATION$_{i,t}$ | | | | 0.884 (1.33) | 0.632 (0.93) |
| GDP$_{i,t}$ | | −0.00012 *** (−3.02) | −0.00013 *** (−3.12) | −0.00012 *** (−3.05) | −0.00013 *** (−3.13) |
| URBAN_RATIO$_{i,t}$ | | −4.923 *** (−4.44) | −4.428 *** (−3.90) | −4.759 *** (−4.29) | −4.357 *** (−3.84) |
| EMP_NONAGR_RATIO$_{i,t}$ | | 0.569 (0.36) | −0.022 (−0.014) | 0.458 (0.29) | −0.06 (−0.03) |
| HEALTH_EXP_RATIO$_{i,t}$ | | −20.057 ** (−2.37) | −20.739 ** (−2.39) | −19.258 ** (−2.26) | −20.08 ** (−2.31) |
| BIRTH_CON_RATE$_{i,t}$ | | −4.923 *** (−3.82) | −4.364 *** (−3.33) | −5.068 *** (−3.91) | −4.509 *** (−3.405) |
| LIFE_EXP$_{i,t}$ | | 0.164 *** (4.69) | 0.169 *** (4.76) | 0.174 *** (4.83) | 0.176 *** (4.83) |
| GOVEXP_EDU_RATIO$_{i,t}$ | | −2.437 (−0.23) | −3.377 (−0.31) | −1.705 (−0.15) | −2.689 (−0.25) |
| ADULT_LIT_RATE$_{i,t}$ | | 0.872 (0.67) | 0.851 (0.65) | 0.892 (0.68) | 0.869 (0.67) |
| RD_EXP_RATIO$_{i,t}$ | | 290.772 *** (4.38) | 305.34 *** (4.55) | 301.22 *** (4.48) | 311.53 *** (4.61) |
| INTERNET_USERS$_{i,t}$ | | −0.026 ** (−2.57) | −0.036 *** (−3.18) | −0.029 *** (−2.81) | −0.038 *** (−3.28) |
| Observations | 1532 (654) | 494 (297) | 494 (297) | 494 (297) | 494 (297) |
| LR Statistic | 378.48 *** | 284.49 *** | 289.15 *** | 286.30 *** | 290.04 *** |
| McFadden R-squared | 0.1810 | 0.4281 | 0.4352 | 0.4308 | 0.4365 |

Note: 1. A positive coefficient on a particular variable implies that the probability of being at the transition stage relative to nascent stage increases when that variable increases by one unit. Standard errors are in parentheses (*** estimated coefficient significantly different from zero, two-tailed test, 1% significance level; ** as above, 5% significance level; * as above, 10% significance level). 2. Estimation I represents the results of the credit union movement. Estimation II represents the results of the credit union movement affected by economic, social, educational, and cultural factors. Estimation III represents the results of the credit union movement affected by economic, social, educational, and cultural factors in the context of the financial crisis of 2008. Estimation IV represents the results of the credit union movement affected by economic, social, educational, and cultural factors around the time that the WOCCU declared the importance of legislative frameworks in 2002. Estimation V represents the results of the credit union movement affected by economic, social, educational, and cultural factors in the context of both the financial crisis and the legislative framework declaration.

**Table 8.** Estimated results of the binary logistic regression (from the transition stage to the mature stage).

| Estimation/Stage | I | II | III | IV | V |
|---|---|---|---|---|---|
| Constant Term | −17.357 *** (−10.79) | −45.29 *** (−3.28) | −44.33 *** (−3.37) | −40.402 *** (−4.78) | −40.504 *** (−4.77) |
| AVG_ASSET$_{i, t}$(Ln) | 0.970 *** (10.27) | 2.300 *** (3.78) | 2.285 *** (3.60) | 2.143 *** (5.96) | 2.146 *** (5.97) |
| LOAN_RATIO$_{i, t}$ | −0.494 (−0.76) | −2.207 (−0.88) | −3.359 (−1.23) | 1.677 (0.75) | 1.657 (0.74) |
| CU_GRO$_{i, t}$ | −0.030 (−0.21) | 0.111 (0.71) | 0.111 (0.69) | 0.083 (0.54) | 0.084 (0.54) |
| MEM_GRO$_{i, t}$ | −1.756 ** (−2.34) | −0.819 (−0.42) | −0.588 (−0.31) | −1.04 (−0.81) | −1.105 (−0.83) |
| CRISIS$_{i, t}$ | | | 1.638 (1.53) | | −0.150 (−0.23) |
| DEREGULATION$_{i, t}$ | | | | 3.855 *** (3.57) | 3.936 *** (3.44) |
| GDP$_{i, t}$ | | −0.00028 (−1.50) | −0.00034 (−1.58) | −0.000001 (−0.58) | −0.000001 (−0.62) |
| URBAN_RATIO$_{i, t}$ | | −19.926 *** (−2.81) | −16.872 ** (−2.35) | −15.983 *** (−5.47) | −16.045 *** (−5.45) |
| EMP_NONAGR_RATIO$_{i, t}$ | | 6.573 (0.996) | 4.192 (0.63) | −3.783 (−0.86) | −3.702 (−0.84) |
| HEALTH_EXP_RATIO$_{i, t}$ | | 136.78 ** (2.48) | 115.07 ** (2.12) | 118.958 *** (4.33) | 119.54 ** (4.32) |
| BIRTH_CON_RATE$_{i, t}$ | | 8.681 (1.56) | 10.531 * (1.65) | 9.848 *** (3.71) | 9.734 *** (3.60) |
| LIFE_EXP$_{i, t}$ | | −0.024 (−0.14) | −0.033 (−0.19) | −0.079 (−0.83) | −0.078 (−0.82) |
| GOVEXP_EDU_RATIO$_{i, t}$ | | −62.681 (−1.61) | −55.023 (−1.43) | | |
| ADULT_LIT_RATE$_{i, t}$ | | 7.688 (1.28) | 7.490 (1.16) | | |
| RD_EXP_RATIO$_{i, t}$ | | 997.93 *** (3.45) | 967.35 *** (3.37) | 888.870 *** (6.27) | 892.38 *** (6.21) |
| INTERNET_USERS$_{i, t}$ | | −0.138 ** (−2.55) | −0.152 *** (−2.71) | −0.125 *** (−4.75) | −0.124 *** (−4.5) |
| Observations | 781 (127) | 337 (40) | 337 (40) | 525 (125) | 525 (125) |
| LR Statistic | 199.84 *** | 170.43 *** | 172.96 *** | 448.34 *** | 448.40 *** |
| McFadden R-squared | 0.2881 | 0.6940 | 0.7044 | 0.7779 | 0.7780 |

Note: 1. A positive coefficient on a particular variable implies that the probability of being at the mature stage relative to the transition stage increases when that variable increases by one unit. Standard errors are in parentheses (*** estimated coefficient significantly different from zero, two-tailed test, 1% significance level; ** as above, 5% significance level; * as above, 10% significance level). 2. Estimation I represents the results of the credit union movement. Estimation II represents the results of the credit union movement affected by economic, social, educational, and cultural factors. Estimation III represents the results of the credit union movement affected by economic, social, educational, and cultural factors in the context of the financial crisis of 2008. Estimation IV represents the results of the credit union movement affected by economic, social, educational, and cultural factors around the time that the WOCCU declared the importance of legislative frameworks in 2002. Estimation V represents the results of the credit union movement affected by economic, social, educational, and cultural factors in the context of both the financial crisis and the legislative framework declaration.

We can see a significant positive coefficient for AVG_ASSET in the binary logistic regression in Tables 7 and 8. As shown in Table 7, the loan ratio, along with the growth ratios of credit unions and their membership, represent the development of credit unions in each country; countries with a high loans-to-assets ratio or a high membership growth ratio might be common phenomena at the nascent stage. Hence, we would expect significant negative coefficients for LOAN_RATIO and MEM_GRO in all estimations. We also found a positive coefficient for CU_GRO in in all estimations, but only Estimation I was significant. However we can see no significant relationship in Table 8. This means that the loan ratio and the membership growth ratio are the most significant negative covariates from the nascent stage to the transition stage, but are insignificant from the transition stage to the mature stage.

For economic indicators, gross domestic product (GDP) has a significant negative relationship only from the nascent stage to the transition stage, as shown in Table 7. However, GDP does not affect the development from the transition stage to the mature stage, as shown in Table 8. The ratio of urban population has significant negative relationship in all estimations, which means that the lower the urban population ratio (URBAN_RATIO), the higher the probability of being at the transition stage relative to the nascent stage, as shown in Table 7. This situation is the same from the transition stage to the mature stage, as shown in Table 8. Even though the non-agricultural employment rate (EMP_NONAGR_RATIO) presents an insignificant positive relationship with the stages of development, indicating that the non-agricultural employment rate does not affect the stages of development.

This research uses health expenditure ratio, birth control rate, and life expectancy as the proxy variables of social indicators. The life expectancy (LIFE_EXP) shows a significant positive relationship with the stages of development in the estimations shown in Table 7: the longer the life expectancy at birth, the higher the probability of being at the transition stage relative to the nascent stage. However, LIFE_EXP does not affect the development from the transition stage to the mature stage, as shown in Table 8. Meanwhile, birth control rate (BIRTH_CON_RATE) has a significant negative relationship with the stages of development in the estimations shown in Table: the lower the reciprocal of the total fertility rate, the higher the probability of being at the transition stage relative to the nascent stage. Conversely, BIRTH_CON_RATE has a significant positive relationship with the estimations from the transition stage to the mature stage, as shown in Table 8. Finally, the health expenditure ratio (HEALTH_EXP_RATIO) presents a significant negative relationship with the stages of development in the estimations shown in Table 7, which means that the lower the health expenditure ratio, the higher the probability of being at the transition stage relative to the nascent stage. However, HEALTH_EXP_RATIO has a significant positive relationship with the estimations from the transition stage to the mature stage shown in Table 8.

For educational indicators, both GOVEXP_EDU_RATIO and ADULT_LIT_RATE have insignificant relationships with the stages of development in the estimations shown in Tables 7 and 8, indicating that the do not affect the stages of development.

Finally, the R&D expenditure ratio shows a significant positive relationship with the stages of development in the estimations shown in Table 7: the higher the R&D expenditure ratio, the higher the probability of being at the transition stage relative to the nascent stage. The situation is the same from the transition stage to the mature stage, as shown in Table 8. The number of Internet users (INTERNET_USERS) has a significant negative relationship with the stages of development in estimations shown in Table 7: the lower the INTERNET_USERS, the higher the probability of being at the transition stage relative to the nascent stage. This situation is the same from the transition stage to the mature stage, as shown in Table 8. When considering the financial crisis (CRISIS) of 2008 and the WOCCU declaration of the importance of legislative frameworks (DEREGULATION) in 2002, only CRISIS shows a significant positive relationship with the stages of development in Table 7: the longer after the financial crisis of 2008, the higher the probability of being at the transition stage relative to the nascent stage. However, CRISIS does not affect the development

from the transition stage to the mature stage, as shown in Table 8. DEREGULATION has an insignificant positive relationship with the probability of being at the transition stage relative to the nascent stage, as shown in Table 7. However, DEREGULATION has a significant positive relationship with the stages of development shown in Table 8: the longer after the WOCCU declaration of the importance of legislative frameworks in 2002, the higher the probability of being at the mature stage relative to the transition stage.

*4.2. Models' Estimation Results*

The empirical results are shown in Table 9, including the coefficient estimation of each model and its adjusted R-squared value. This research adopted pooled panel data to estimate the fixed-sample effects, and the Hausman test was also used to examine the random and fixed effects of the models. Since the results were all significant, this study chose the fixed-effects method to conduct the estimation.

A positive coefficient on a particular variable implies that the penetration rate increases when that variable increases by one unit. Model I represents the results of the credit union movement in the context of the financial crisis and the legislative framework declaration. Model II represents the results of the credit union movement affected by economic factors. Model III represents the results of the credit union movement affected by economic and social factors. Model IV represents the results of the credit union movement affected by economic, social, and educational factors. Model V represents the results of the credit union movement affected by economic, social, educational, and cultural factors.

Table 6 shows a significant positive coefficient for AVG_ASSET in Models I, II, and III, which means that the greater the average assets of the credit union (AVG_ASSET), the higher the penetration rate.

As shown in Table 9, the loan ratio, along with the growth ratios of the credit unions and their membership, represents the development of credit unions in each country; high loans-to-assets ratios or a high growth ratios of credit unions or their members might be common phenomena in countries with high penetration rates. Thus, we would expect significant positive coefficients for LOAN_RATIO in Models I, II, and III, for CU_GRO in Models IV and V, and for MEM_GRO in Model V. The financial crisis (CRISIS) of 2008 shows a significant positive relationship in Models I, II, III, and IV: the longer after the financial crisis of 2008, the higher the penetration rate. The WOCCU declaration of the importance of legislative frameworks (DEREGULATION) in 2002 shows a significant negative relationship in Model I: the longer after the WOCCU declaration of the importance of legislative frameworks in 2002, the lower the penetration rate. If we consider the economic, social, and educational factors in Model IV, it shows the opposite (significant positive) relationship.

For economic factors, GDP has a significant negative relationship in Models II and III, indicating that the higher the GDP, the lower the penetration rate. The ratio of urban population (URBAN_RATIO) has a significant negative relationship in Models II and III, which means that the higher the URBAN_RATIO, the lower the penetration rate. However, the non-agricultural employment rate (EMP_NONAGR_RATIO) presents an insignificant positive relationship, indicating that EMP_NONAGR_RATIO does not affect the penetration rate, as shown in Table 6.

For social factors, the health expenditure ratio (HEALTH_EXP_RATIO) presents an insignificant relationship, indicating that HEALTH_EXP_RATIO does not affect the penetration rate. The birth control rate (BIRTH_CON_RATE) has a significant positive relationship in Models III and V: the higher the reciprocal of the total fertility rate, the higher the penetration rate. Finally, life expectancy (LIFE_EXP) shows an insignificant relationship in Models III and IV. If we consider the cultural factors in Model V, then it shows the opposite (significant negative) relationship: the longer the life expectancy at birth, the lower the penetration rate.

**Table 9.** Models' estimation results.

| Models | I | II | III | IV | V |
|---|---|---|---|---|---|
| Constant Term | −0.06 (0.046) | 0.221 *** (0.075) | 0.111 (0.096) | −0.129 ** (0.062) | 0.172 ** (0.082) |
| Covariate/PENETRATION | | | | | |
| AVG_ASSET$_{i,t}$(Ln) | 0.0132 *** (0.003) | 0.014 *** (0.003) | 0.014 *** (0.003) | 0.00005 (0.0019) | −0.0056 ** (0.002) |
| LOAN_RATIO$_{i,t}$ | 0.057 ** (0.023) | 0.059 *** (0.023) | 0.063 *** (0.024) | 0.021 (0.013) | 0.022 (0.014) |
| CU_GRO$_{i,t}$ | 0.002 (0.0015) | 0.002 (0.0014) | 0.002 (0.0014) | 0.0015 *** (0.0006) | 0.0012 ** (0.0005) |
| MEM_GRO$_{i,t}$ | 0.003 (0.0028) | 0.003 (0.0026) | 0.003 (0.0026) | −0.00017 (0.0013) | 0.0041 ** (0.0016) |
| CRISIS$_{i,t}$ | 0.031 *** (0.008) | 0.053 *** (0.007) | 0.046 *** (0.008) | 0.019 *** (0.0046) | 0.0063 (0.0054) |
| DEREGULATION$_{i,t}$ | −0.021 *** (0.008) | 0.00012 (0.008) | −0.009 (0.009) | 0.012 * (0.006) | −0.01 (0.008) |
| GDP$_{i,t}$ | | −0.000003 *** (0.0000007) | −0.000003 *** (0.0000007) | −0.00000017 (0.0000008) | −0.0000006 (0.0000008) |
| URBAN_RATIO$_{i,t}$ | | −0.657 *** (0.135) | −0.76 *** (0.139) | 0.078 (0.094) | −0.136 (0.114) |
| EMP_NONAGR_RATIO$_{i,t}$ | | 0.061 (0.046) | 0.049 (0.046) | 0.011 (0.021) | 0.038 (0.025) |
| HEALTH_EXP_RATIO$_{i,t}$ | | | 0.21 (0.306) | −0.235 (0.199) | 0.283 (0.265) |
| BIRTH_CON_RATE$_{i,t}$ | | | 0.223 *** (0.08) | 0.045 (0.046) | 0.105 ** (0.045) |
| LIFE_EXP$_{i,t}$ | | | 0.0007 (0.096) | 0.0003 (0.0005) | −0.0016 ** (0.0006) |
| GOVEXP_EDU_RATIO$_{i,t}$ | | | | 1.026 *** (0.228) | 0.985 *** (0.269) |
| ADULT_LIT_RATE$_{i,t}$ | | | | 0.054 (0.061) | −0.0017 (0.074) |
| RD_EXP_RATIO$_{i,t}$ | | | | | 0.663 (1.317) |
| INTERNET_USERS$_{i,t}$ | | | | | 0.0007 *** (0.0002) |
| Observations | 1659 (113) | 1480 (101) | 1443 (98) | 765 (69) | 534 (52) |
| F-Test | 80.29 *** | 54.17 *** | 50.15 *** | 50.72 *** | 34.8 *** |
| Cross-Section F-Test | 53.41 *** | 58.21 *** | 48.23 *** | 15.3 *** | 13.41 *** |
| Hausman Test | - | 97.49 *** | 64.92 *** | 24.78 ** | 55.37 *** |
| R-squared | 0.1625 | 0.2627 | 0.2881 | 0.2221 | 0.2933 |
| Adjusted R-squared | 0.1594 | 0.2582 | 0.2821 | 0.2076 | 0.2715 |

Note: 1. A positive coefficient on a particular variable implies that the penetration rate increases when that variable increases by one unit. Standard errors are in parentheses (*** estimated coefficient significantly different from zero, two-tailed test, 1% significance level; ** as above, 5% significance level; * as above, 10% significance level.). 2. Model I represents the results of the credit union movement in the context of the financial crisis and the legislative framework declaration. Model II represents the results of the credit union movement affected by economic factors. Model III represents the results of the credit union movement affected by economic and social factors. Model IV represents the results of the credit union movement affected by economic, social, and educational factors. Model V represents the results of the credit union movement affected by economic, social, educational, and cultural factors.

For educational factors, the government expenditure on education (GOVEXP_EDU_RATIO) has a significant positive relationship in Models IV and V, which means that the higher the GOVEXP_EDU_RATIO, the higher the penetration rate. Meanwhile, adult literacy rate (ADULT_LIT_RATE) has an insignificant relationship in Models IV and V, which indicates it does not affect the penetration rate.

Finally, for cultural factors, only Internet users (INTERNET_USERS) shows a significant positive relationship, in Model V: the higher the INTERNET_USERS, the higher the penetration rate. The R&D expenditure ratio (RD_EXP_RATIO) does not have a significant relationship in Model V, which indicates that RD_EXP_RATIO does not affect the penetration rate.

## 5. Conclusions

The aim of this paper was to find a development framework for the analysis of credit union movements by grouping credit unions into different category types. Within the heterogeneous reality of the worldwide credit union movement, the typology provides a clearer understanding of the dynamics of change and development. It also provides a useful tool for analyzing this dynamic category by offering a classification of credit unions' development.

We found the key covariates of influence when analyzing the original typology to add further explanation of the development of credit union movements. Only the positive influence of the asset scale, financial crisis, legislative framework, life expectancy, and R&D expenditures covariates, and the negative influence of the urban population ratio and birth control rate covariates, are significant from the nascent stage to the mature stage. This supports the notion that the stage of development of credit union movements depends on asset scale, financial crisis, legislative framework, economy, society, and culture in different countries.

We also found that loan ratio and the growth ratio of membership are significant negative covariates from the nascent stage to the transition stage, but are insignificant from the transition stage to the mature stage. For economic indicators, gross domestic product has a significant negative relationship only from the nascent stage to the transition stage, and does not affect the development from the transition stage to the mature stage. For social indicators, life expectancy shows a significant positive relationship for the transition stage relative to the nascent stage, and does not affect the development from the transition stage to the mature stage. Although birth control rate and health expenditure ratio both have significant negative relationships with development from the nascent stage to the transition stage, they have significant positive relationships with development from the transition stage to the mature stage. For cultural indicators, the number of Internet users has a significant negative relationship in the transition stage relative to the nascent stage. This situation is the same for development from the transition stage to the mature stage.

When considering the financial crisis of 2008 and the WOCCU declaration of the importance of legislative frameworks in 2002, only CRISIS shows a significant positive relationship with development from the nascent stage to the transition stage. However, it does not affect the development from the transition stage to the mature stage. Meanwhile, DEREGULATION has an insignificant positive relationship with development from the nascent stage to the transition stage, but it has a significant positive relationship with the probability of being in the mature stage relative to the transition stage.

The point was made that a key element determining whether a credit union industry is nascent, or has reached a higher level of development, is its asset scale. In the United States, Canada, and Australia, the historical asset scale from 1995 to 2015 is universally recognized. The movement of Ireland was categorized as being in transition in 1996, then moved forward to the mature stage in 2003. In another example, the movement of Great Britain was categorized as nascent in 1996, then moved forward to the transition stage in 2000. This growth of asset scale may be a reflection of deregulation, professionalization of management, diversification of products and services, and emphasis on economic viability

and long-term sustainability. In the United States, Canada, and Australia, the historical penetration from 1995 to 2015 is universally recognized. The movement of Thailand was categorized as nascent in 1996, and then in transition in 1997, later moving forward to the mature stage in 2003. In another example, the movement of Brazil was categorized as being in transition in 1996, and then moved forward to the mature stage in 2006. The penetration may be a reflection of loose common bonds, the electronic technology environment, and well-organized, progressive trade bodies.

We also found from robustness tests that the positive influence of the asset scale, loan ratio, credit union growth, financial crisis, birth control rate, government expenditure on education, and Internet users covariates, and the negative influence of the gross domestic product (GDP) and urban population ratio covariates, are significant to the penetration rate of each country. This supports the notion that the penetration rate of credit union movements depends on the asset scale, loan ratio, credit union growth, financial crisis, economy, society, education, and culture of the country. This lends support to the recognition of the diversity of the credit unions' development.

Rich (1992) found that classification systems are judged not by the ease or neatness through which the organizations are grouped, but by their utility and their ability to replicate reality in the final analysis. An organizational typology that does not adequately and recognizably reflect the world that it purports to categorize has little value, either in terms of usefulness or as a model of reality. The usefulness and limitations of the typology are neatly affected by the observation.

In the future, there are still many factors influencing the credit union movements in these countries that need to be verified. These factors trace the evolution of the credit union movements, where issues related to efficiency, technology adoption, product diversification, mergers, failures, and equities need to be considered as goals of the credit unions, while also weighing interest rate regulations, common bond requirements, taxation, deposit insurance, and capital regulation, as well as other topics in the regulatory environment.

**Author Contributions:** Conceptualization, C.-M.K.; Data Curation, C.-M.K.; Formal Analysis, C.-M.K.; Funding Acquisition, C.-M.K.; Investigation, C.-M.K.; Methodology, C.-M.K.; Project Administration, C.-M.K.; Resources, M.-C.W. and L.L.; Software, L.L.; Supervision, M.-C.W. and L.L.; Validation, M.-C.W. and L.L.; Visualization, L.L.; Writing—Original Draft, C.-M.K.; Writing—Review and Editing, L.L. All authors have read and agreed to the published version of the manuscript.

**Funding:** This research received no external funding.

**Data Availability Statement:** Dataset available at http://www.woccu.org/publications/statreport, accessed on 24 November 2017.

**Acknowledgments:** The authors thank the editor and three anonymous referees for their helpful comments and suggestions.

**Conflicts of Interest:** The authors declare no conflict of interest.

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
