# Peer review of "The Dynamic Typology in the Development Process of Credit Union Movements"

_ijfs, doi:10.3390/ijfs10020029_

Round 1

Reviewer 1 Report

The authors have not incorporated the my first round comments into the paper they have submitted. Unless they do, the paper is not acceptable for publication.

Reviewer 2 Report

The framework for this research is that of Sibbald, Ferguson, and McKillop (Annals of Public and Cooperative Economics, 2002).  Three stages of development are posited, nascent, transition, and mature. These authors argue that the progression between stages of development is driven by factors of situational leadership, the complexion of trade associations, professionalization, regulatory and legislative initiatives, and technology.  I appreciate that the authors of the current research paper seek to elevate the analysis by employing data and rigorous econometrics models.

The dependent variable for the ordered logistics model is the classification of credit unions (e.g., nascent, transition, mature).  A better description of this variable is needed.  As best I can determine, the variable is taken from a report issued by the Commission on Credit Unions in 2012 (lines 179-182). I could be wrong about this since the source of the variable is not clearly stated in the manuscript.  Critically, the authors should make clear whether this variable is updated annually.  About twenty years of data are used to conduct the empirical analysis.  If the country assignments as nascent, transition, and mature are not updated annually, then it is unclear why 20 years of data are used.  Why not just conduct analysis using information for the time period when the classifications were set? In the conclusion section, the authors note changes in type for several countries.  If the classifications are updated annually, then it would be useful to explain that process.  I reviewed the cited data source: woccu.org and found the annual statistics on the independent variables, but no data on credit union type. 

Presumably, the organization/author that created the dependent variable used factors such as asset size, loan ratio, member growth, regulation, etc. That is, the variables in the ordered logistic regression presented in this manuscript are those that define the classifications and that would have been used by the organization/author to assign the classifications.  What then is the goal of estimating the logistic model?  Is it to reverse engineer the formula that was used to classify credit union regimes?  If so, then several interesting questions might be addressed.  First, does the logistic model imply that any country is misclassified?  Second, which countries does the logistic model suggest are on the verge of migrating to a higher level?  I struggle to understand the value of demonstrating the significance of the independent and proxy variables when they are just the definition of the dependent variable. To derive value, it seems that the estimated models need to be applied in some way, such as answering the two questions posed above. 

Are the factors that drive the transition from a nascent state to a transition state the same as for credit unions moving from transition to mature? If not, should the empirical work focus on the transition phases separately (e.g., from nascent to transition and from transition to mature)? 

The empirical models include “independent” financial variables on credit unions: average assets, loan-asset ratio, CU growth, and member growth.  Given the number of observations, this data does not appear to be microeconomic information but aggregated information for all credit unions in each country.  Please clarify if these variables represent industry averages, median, or something else. 

Round 2

Reviewer 1 Report

the paper is satisfactory in this final second revision

Reviewer 2 Report

Authors have responded to comments from the last review. 

This manuscript is a resubmission of an earlier submission. The following is a list of the peer review reports and author responses from that submission.